# Sensorimotor integration enhances temperature stimulus processing

**Lindsay S. Anderson**[1,2], **Jamie D. Costabile**[1¤], **Sina Schwinn**[1], **Delia Calderon**[3,4], **Martin Haesemeyer** [1]*

**1** Department of Neuroscience, The Ohio State University College of Medicine, Columbus, Ohio, United States of America, **2** Neuroscience Graduate Program, The Ohio State University, Columbus, Ohio, United States of America, **3** Nationwide Children's Hospital, Columbus, Ohio, United States of America, **4** Molecular, Cellular and Developmental Biology Program, The Ohio State University, Columbus, Ohio, United States of America

¤ Current address: Hitachi Solutions America, Ltd., Irvine, California, United States of America
* haesemeyer.1@osu.edu

**Data availability statement:** All code used in this study is available in the following repositories: Tail tracking of freely swimming

## Abstract

Animals optimize behavior by integrating sensory input with motor actions. We hypothesized that coupling thermosensory information with motor output enhances the brain's capacity to process temperature changes, leading to more precise and adaptive behaviors. To test this, we developed a virtual "thermal plaid" environment where zebrafish either actively controlled temperature changes (sensorimotor feedback) or passively experienced the same thermal fluctuations. Our findings demonstrate that sensorimotor feedback amplifies the influence of thermal stimuli on swim initiation, resulting in more structured and organized motor output. We show that previously identified mixed-selectivity neurons that simultaneously encode thermal cues and motor activity enable the integration of sensory and motor feedback to optimize behavior. These results highlight the role of sensorimotor integration in refining thermosensory processing, revealing critical neural mechanisms underlying flexible thermoregulatory behavior. Our study offers new insights into how animals adaptively process environmental stimuli and adjust their actions, contributing to a deeper understanding of the neural circuits driving goal-directed behavior in dynamic environments.

## Author summary

As animals explore their environment, they constantly encounter new sensory information. Some sensory inputs come from their own movements, such as sounds while walking across leaves while others indicate a charging predator. Animals therefore possess strategies to distinguish self-generated sensory feedback from sensory stimuli generated by their surroundings. In some cases, brains filter out self-generated stimuli as evidenced by our inability to tickle ourselves. Under other circumstances, information can be gained by understanding the relationship between stimuli and behavior.

fish: https://bitbucket.org/jcostabile/tracking. Live tracking and laser control setup "Zebratrack": https://github.com/haesemeyer/ZebraTrack. Data analysis: https://github.com/haesemeyer/plaid_pub. Raw experimental data has been deposited to DANDI in NWB format: Plaid Experiments https://doi. org/10.48324/dandi.000888/0.241014.2127. Replay Experiments https://doi.org/10.48324/ dandi.000889/0.241014.2127. Plaid hyperparameter set https://doi.org/10.48324/ dandi.000485/0.241014.2127. Replay hyperparameter set https://doi.org/10.48324/ dandi.000486/0.241014.2127. The analysis code was written while transitioning to using NWB for behavioral data. The code therefore does not load the NWB files deposited at DANDI but instead loads the raw experimental data files created by our setup. These are deposited at: Plaid Experiments https://doi.org/10.5281/zenodo.13930780. Replay Experiments https://doi.org/10.5281/zenodo.13935291. The stimulus/behavior input data for predicting mixed selectivity neuron responses is deposited at: https://doi.org/10.5281/zenodo.13935648.

**Funding:** Research reported in this publication was supported by the National Institute Of Neurological Disorders And Stroke of the National Institutes of Health under Award Number R01NS123887 to MH and by the Office Of The Director, National Institutes of Health under Award Number R24OD037693 to MH. The funders had no role in study design, data collection and analysis, decision to publish, or preparation of the manuscript. MH, LA and JDC received salary from the funders. The content is solely the responsibility of the authors and does not necessarily represent the official views of the National Institutes of Health.

**Competing interests:** I have read the journal's policy and the authors of this manuscript have the following competing interests. JDC is employed by Hitachi Solutions. The other authors declare that no competing interests exist.

Here, we investigated whether larval zebrafish utilize sensory feedback during behavioral thermoregulation. We compared the relationship of thermosensory stimuli and behavior under two conditions: One where larval zebrafish navigated thermal gradients and one where they passively experienced changes in temperature. In the first condition, changes in temperature are linked to behavior. We found that this thermosensory feedback increased the behavioral response of larval zebrafish to temperature stimuli. As animals thermoregulate by seeking out comfortable temperatures, this suggests that detecting a link between behavior and temperature change will enhance the reaction to temperature stimuli. This could serve the goal of increasing thermoregulatory behaviors in an environment where they can be used efficiently.

## Introduction

Integrating sensory information with information about one's own behavioral actions provides valuable insight into the state of the environment. Perception is therefore considered an active rather than passive process in which animals attempt to gather information through behavior [1]. For example, relating sound cues to changes in our walking pattern can tell us if the footsteps we are hearing are likely our own or not. In goal-directed behaviors, sensory feedback is critical to judge the success of behavioral actions. Behavioral thermoregulation is a goal-directed program in which animals seek external temperatures that allow them to maintain optimal body temperature. Behavioral thermoregulation is ubiquitous in animals regardless of complexity, as basic cellular processes strongly depend on temperature [2–4]. Ectothermic invertebrates such as *C. elegans* and *Drosophila* readily navigate temperature gradients to thermoregulate [5–8] as do ectothermic vertebrates such as toads and zebrafish [9–12]. However, behavioral thermoregulation is also conserved in mammals, including humans [13–15] despite their ability to autonomously regulate body temperature, due to the energetic cost of autonomous temperature regulation [16,17].

Like any regulatory task, behavioral thermoregulation should benefit from evaluating the sensory feedback generated by ongoing behaviors. However, it is unknown whether animals integrate information about behavioral actions with thermosensory feedback to optimize thermoregulatory behavior; e.g., coincidence of movements and perceived changes in temperature could inform animals about the presence of thermal gradients, while integrating information about travel distance with temperature change could be used to infer the gradient's slope. Absence of a link between changes in temperature and movement would on the other hand signal environmental factors such as changes in cloud cover which the animal cannot exploit for thermoregulation. Here, we address whether animals use thermosensory feedback of behavior in the vertebrate model larval zebrafish. Larval zebrafish thermoregulate by modulating swim rates and turn kinematics to navigate temperature gradients [11,12]. Both brainstem and forebrain circuits involved in controlling this behavior have been previously identified [12,18]. This highlights the distributed nature of the neural mechanisms that coordinate motor output during thermal navigation. Through whole-brain imaging coupled with a new analysis method, we recently identified mixed-selectivity neurons within the larval zebrafish brain that display linear and nonlinear integration of thermosensory and behavioral information [19]. This suggests that the larval zebrafish brain integrates information about behavioral actions with thermosensory feedback. Since thermosensory feedback can be used for operant conditioning in larval zebrafish [20,21] these neurons might exclusively be a substrate for operant learning or could serve a role in adjusting ongoing behavior based on sensory feedback.

To test if larval zebrafish integrate thermosensory information with behavioral actions to adjust thermoregulation we used a laser-tracking setup to generate a series of small temperature gradients arranged in a plaid pattern. Zebrafish either navigated this virtual "thermal plaid" or experienced it through playback as a yoked control. This allowed us to compare thermosensory receptive fields under two conditions. In the first condition, movements of the animal lead to changes in temperature, which means that each behavioral action generated sensory feedback ("sensorimotor feedback"). In the second condition, changes in temperature are decoupled from behavior, breaking the feedback loop. Using artificial neural network models, we extracted the thermosensory receptive field as well as the influence of behavioral history on swim initiation in both conditions. Our analysis suggests that behavior becomes more predictable as its temporal structure becomes more coherent (i.e., structured output) and that the stimulus exerts a larger influence on behavior during sensorimotor feedback. At the same time, predictive models of mixed-selectivity neurons suggest that these neurons encode the integration of thermosensory stimuli with ongoing behavior to facilitate the detection of sensorimotor feedback. In summary, our results show that larval zebrafish integrate thermosensory information with information about ongoing behavior to modulate thermoregulatory behavior.

## Results

### A paradigm to probe sensorimotor integration

To probe the role of sensorimotor feedback on thermoregulatory behaviors we designed a paradigm in which larval zebrafish navigate a virtual thermal environment. Larval zebrafish swim in discrete bouts that are on average 100-200 ms long [26]. These swim bouts are separated by stationary periods, the interbout intervals which last 500-1800 ms. We therefore focused our paradigm on modulating the amount of sensory feedback larval zebrafish would receive during the swim periods versus the intervening interbout periods. To accomplish this we used an infrared laser guided by online tracking [23] (Fig 1A). This setup allowed us to present thermal gradients as a fish might experience in nature by modulating temperature according to fish position, as well as presenting changes in temperature that are decoupled from position which would be difficult to obtain in a gradient setup. Using this setup, we modulated the temperature of larval zebrafish according to one of two rules (Fig 1B). In the first case, temperature was coupled to the position of the fish within a 10 cm sized arena in the form of a "thermal plaid", forming multiple mini-gradients ranging from 28 °C at the warmest point to 25 °C at the coolest over a distance of 21 mm ("Plaid" condition). We chose a plaid-like temperature profile for two reasons. Within the plaid, fish will experience both increases and decreases in temperature even for short movement distances and unlike a circular gradient, there is no defined relationship between temperature and distance to the edge of the dish which avoids confounds introduced by larval zebrafish hugging the edge of the arena (thigmotaxis [32]). Fish in the second group explored the same 10 cm arena, however their temperature was controlled by the temperature experienced by a corresponding fish within the first group ("Replay" condition). This difference in the relationship of the stimulus to the location within the arena (Fig 1C) resulted in a difference in sensorimotor feedback between the two conditions, while maintaining the sensory stimulus itself. In the Plaid condition, changes in temperature were coupled to movements enacted by the fish. In the Replay condition on the other hand changes in temperature were uncoupled from behavior (Fig 1D). From the fish's perspective, the largest change in temperature should occur while moving (i.e., during swim bouts) in the Plaid condition, while temperature changes should be evenly distributed across bout and interbout periods in the Replay condition. This is indeed what we found. The

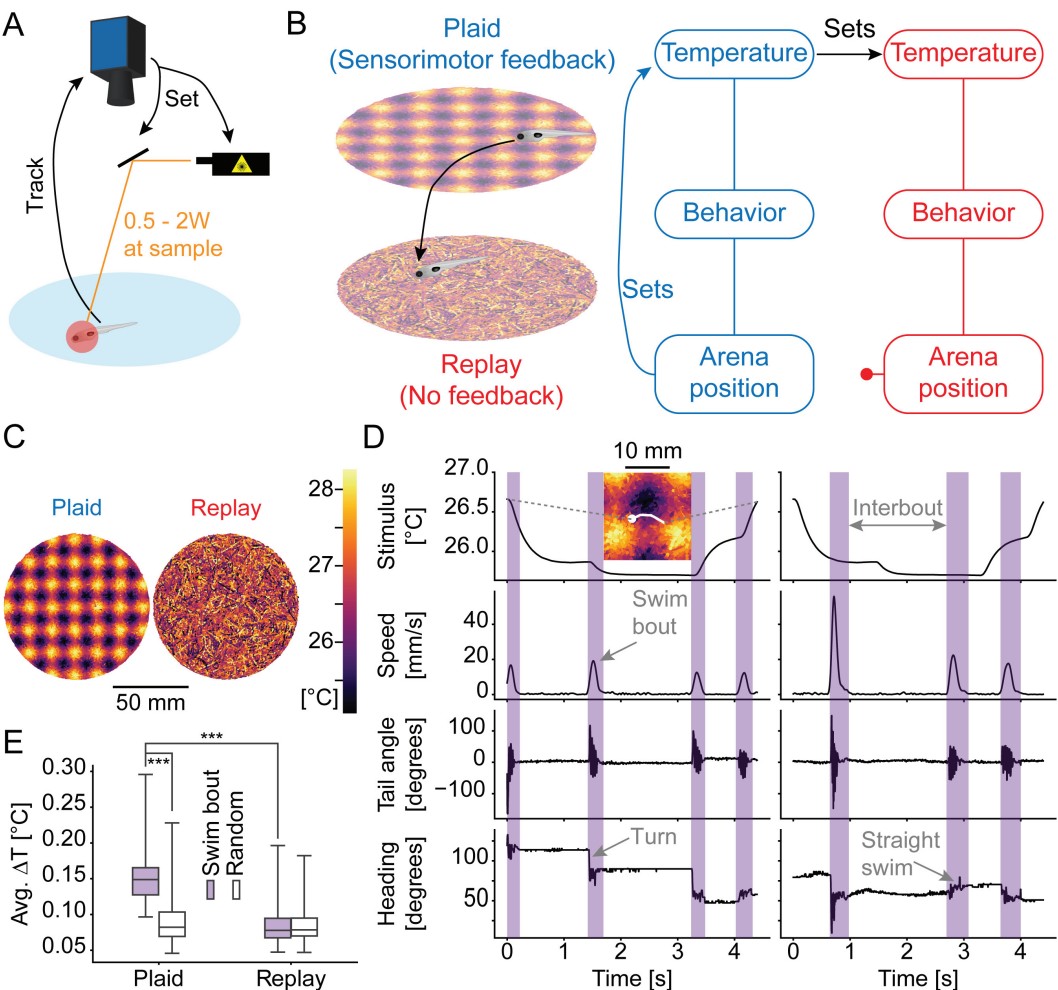

**Fig 1. A paradigm to test the influence of sensorimotor integration on sensory processing. A)** Illustration of the behavioral setup in which fish are continuously tracked and their temperature is modulated via an infrared laser that is centered on the fish's head at all times. **B)** Illustration of the behavioral paradigm. **C)** Relationship of arena position of the fish to temperature in Plaid (left) and Replay (right) conditions across experiments. **D)** Illustration of extracted swim bout kinematics and 4.5 s long example trace for a pair of Plaid (left) and Replay (right) experiments showing delivery of the same stimulus but change in sensorimotor feedback. Stimulus temperature top, swim speed in second row revealing structure of larval zebrafish movement as periods of rests with intermittent swim bouts, third row reveals tail movements that are used as sensitive indicators to delineate the starts of swim bouts, bottom row shows fish heading within the arena revealing turns and straight swims; swim periods are marked in purple. We note that the rapid alternations in heading during swim-bouts are caused by the tail beats of the fish. Inset in Plaid stimulus trace shows an enlarged view of the pattern depicted in C with the example movement depicted across D overlaid as a white line. **E)** For Plaid and Replay conditions the average temperature change experienced by larval zebrafish during swim-bouts (filled purple boxes) and during random time-intervals of the same length (open boxes) within each experiment. Comparison of Plaid within swim-bout vs. random: Rank Sum test, p-value<0.001; statistic = -7.81; N = 52 fish. Comparison of Plaid within swim-bout vs. Replay within swim-bout: Rank Sum test, p-value<0.001; statistic = -7.98; N = 52 fish.

paradigm led to a median temperature change of 0.15 °C for each swim bout in the Plaid condition, while the median temperature change was 0.08 °C in the Replay condition. Importantly, within comparable time-intervals during inter-bout periods the change in temperature in either condition was 0.08 °C (Fig 1E). That is, the temperature change experienced during swims was higher than during stationary periods in the Plaid condition, whereas there was no

difference during Replay conditions. Although these changes are small, larval zebrafish use differences on the order of 0.1 °C to modulate behavior during gradient navigation [12,23]. This paradigm allowed us to assess whether larval zebrafish change the processing of thermosensory stimuli dependent on the presence or absence of sensorimotor feedback.

## Sensorimotor feedback increases the stimulus influence on behavior

When larval zebrafish are deprived of visual feedback during behavior, they strongly suppress motor output, entering a state of learned helplessness [33]. A reason for this might be that when animals move through the world, they generally experience optic flow [34] and e.g., rodents will compute mismatch signals when this expectation is violated [35,36]. The complete absence of visual feedback may therefore signal that behavior is futile since it appears to the animal as if it is not moving. However, whether or not movements lead to temperature changes is strongly dependent on environmental conditions. Within a temperature gradient any swim likely changes the temperature of the animal while swimming in a constant temperature pool does not. The expectation of thermal feedback with movement is therefore conceivably weaker than the expectation of visual feedback. In line with this we found no gross changes in behavioral output based on stimulus coupling as assessed by comparing swim kinematics across the Plaid and Replay conditions. While there was a reduction in interbout intervals less than 1 s in the Replay condition the overall distributions were not significantly different (Fig 2A). Distributions of swim distances and angles turned per swim were highly similar in the Plaid and Replay conditions indicated by the large overlap in distributions (Fig 2B and 2C). Importantly, in both the Plaid and Replay condition, interbout intervals were shorter and swim distances were longer than during baseline conditions in which fish were swimming at a constant temperature of 25 °C (Fig 2A and 2B). This indicates that the higher temperature increased swim vigor under both conditions. It is therefore unlikely that the absence of thermal feedback in the Replay condition induced learned helplessness.

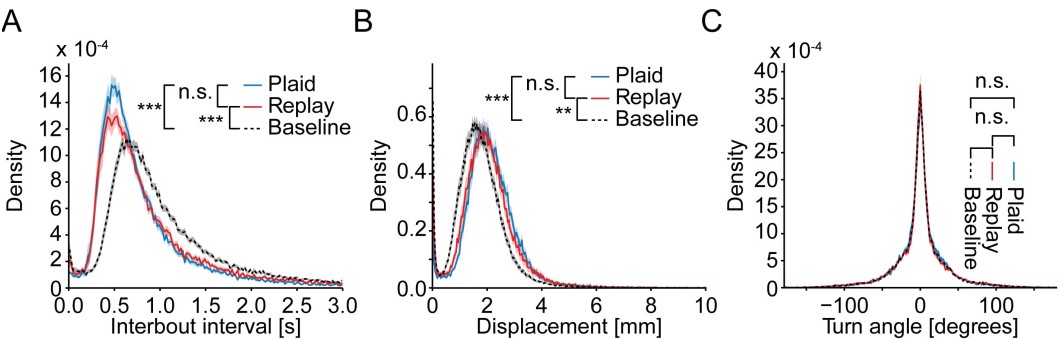

**Fig 2. Absence of feedback does not induce learned helplessness. A)** Density of interbout intervals across Plaid (blue) and Replay (red) experiments during the stimulus phase as well as combined data during the baseline phase (black dashed). Statistics are based on a boot-strapped KS test see Methods. Plaid vs. Replay: p = 0.2548; statistic = 0.0486; N = 104 experiments. Plaid vs. Baseline: p<0.001; statistic = 0.2283; N = 156. Replay vs. Baseline: p<0.001; statistic = 0.1897; N = 156. **B)** Density of per-swim displacements across Plaid (blue) and Replay (red) experiments during the stimulus phase as well as combined data during the baseline phase (black dashed). Plaid vs. Replay: p = 0.218; statistic = 0.0664; N = 104. Plaid vs. Baseline: p<0.001; statistic = 0.1975; N = 156. Replay vs. Baseline: p = 0.002; statistic = 0.1362; N = 156. **C)** Density of swim turn angles across Plaid (blue) and Replay (red) experiments during the stimulus phase as well as combined data during the baseline phase (black dashed). Plaid vs. Replay: p = 0.9003; statistic = 0.0121; N = 104. Plaid vs. Baseline: p<0.067; statistic = 0.0362; N = 156. Replay vs. Baseline: p = 0.109; statistic = 0.0338; N = 156. In all panels shaded error region demarcates bootstrap standard error within each bin.

The increased prevalence of the shortest interbout intervals in the Plaid over the Replay condition (Fig 2A) suggested that sensory feedback enhances swim generation. To understand the basis of this change we were specifically interested in testing for differences in sensory processing and structuring of the behavior. To quantify these two aspects, we sought to extract sensory and behavioral receptive fields. The former describes the transformation of thermal stimuli into swim generation, while the latter quantifies aspects like refractory periods or the influence of swim history on swim initiation. We had previously extracted such receptive fields using white-noise stimuli and generalized linear models (GLM) which revealed the importance of temperature change in guiding swim initiation and which identified a clear refractory period after swimming [23]. Since white-noise stimuli are random and by construction controlled by the experimenter rather than the animal, we could not use them for our current purpose since they would lack sensorimotor feedback. In the current experiments, where larval zebrafish controlled the temperature stimulus, we instead faced the challenge of long stimulus autocorrelation times (S1 FigA) which makes it challenging to extract receptive fields. We therefore used a method based on a convolutional neural network (CNN) we recently developed, Model Identification of Neural Encoding (MINE) [19]. This approach allows us to recover receptive fields even for stimuli with high autocorrelation times, by fitting a CNN relating the stimulus to the response and subsequently using Taylor expansion to extract the receptive field [19]. We previously demonstrated that this approach requires less tuning than receptive field computations based on regularized linear regression and generally yields a more faithful representation of the true receptive field [19]. Here, we designed a CNN to predict the probability of performing a swim bout based on both sensory and behavioral history (Fig 3A). This network received the following inputs as predictors: A one-second history of past temperatures experienced by the fish and a one-second history of previous swim bouts. These inputs were chosen since we and others had previously shown that they influence swim behavior during thermal stimulation and exploration [23,37].

We trained the CNN on 80% of the data after optimizing some of the hyper-parameters on a separate dataset (see Methods and S1B and S1C Fig). To account for potential artifacts due to the long stimulus autocorrelation times, we generated a control condition by rotating the network outputs (generated bouts) with respect to the inputs in the training data. The idea was that this control would allow us to later estimate the noise-floor of extracted receptive fields. After training, we measured how well the networks classified individual frames into those that contained a swim-bout vs. those that did not. Summarizing the results using the area under the receiver-operator-curve (ROC-AUC analysis) showed that in 68% of cases the networks ranked a randomly selected frame with a swim-bout higher than one without (Figs 3B and S1D). This approaches the performance of the GLMs we previously fit on white-noise stimulus data which did so in 71% of cases [23]. Interestingly, the prediction was slightly worse for the model fit on Replay data, which might suggest that behavior is less predictable during the absence of sensorimotor feedback (Fig 3B).

We subsequently extracted receptive fields as a compact representation of how inputs affect the behavioral outputs by differentiating the networks [19]. Here, these receptive fields described the influence of temperature and swim-bout history across time on the generation of the current swim. The thermosensory receptive field (Fig 3C) had a similar structure under both Plaid and Replay conditions, however sensory coupling significantly increased the magnitude of the coefficients (S2 FigA). As expected based on previous results [23], swim initiation was guided by both absolute temperature and changes in the temperature stimulus as evidenced by the mixture of positive and negative coefficients in the receptive fields. Since the coefficients 600 ms before swim initiation were considerably more negative in the Plaid condition (Fig 3C), we expect that sensorimotor feedback enhances the sensitivity to

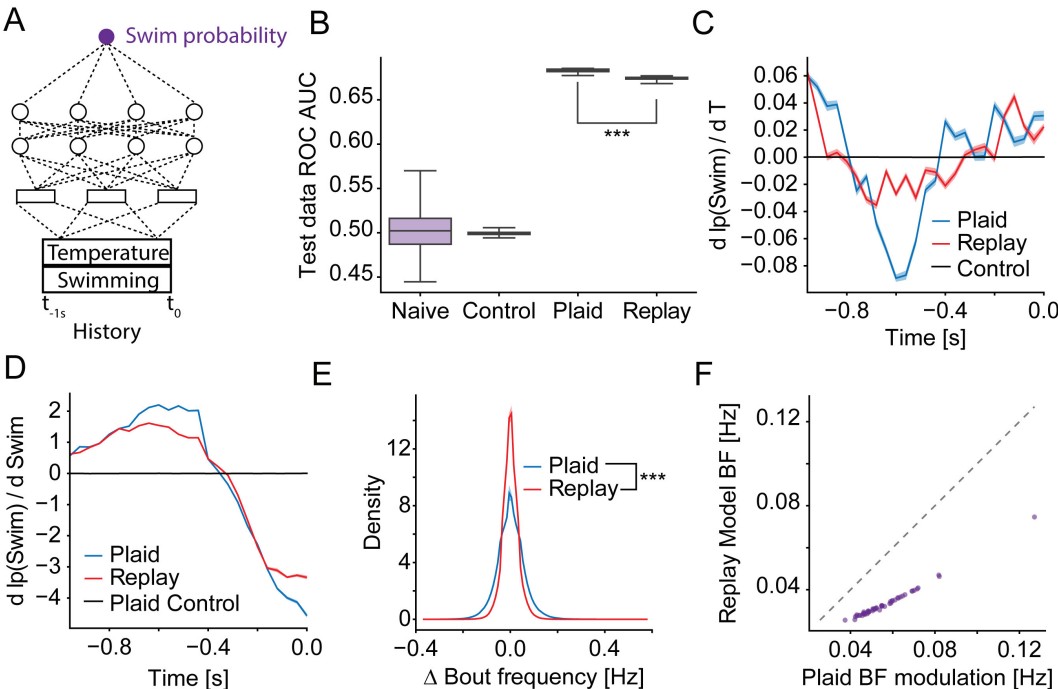

**Fig 3. Coupling of sensation and action modulates receptive fields and increases stimulus influence. A)** Illustration of the convolutional neural network that uses the Temperature stimulus experienced over the last 10 seconds and the swim history to predict the probability of swimming. **B)** Classifier performance as Area under the receiver-operator-curve (ROC-AUC) of classifying individual frames into swim and non-swim frames for naive networks and those trained on circularly permuted data as well as experimental data (N=50 iterations). Comparison of Plaid vs. Replay ROC AUC: Rank Sum test, p<0.001; statistic = -8.62; N = 50 iterations. **C)** Coefficients of the thermosensory receptive field versus time before a swim-bout. Model fit on Plaid data (blue), Replay data (red) and control data (black). Errors are bootstrap standard errors across model fits (N = 50 iterations). **D)** Coefficients of the bout history receptive field versus time before a swim-bout. Model fit on Plaid data (blue), Replay data (red) and control data (black). Errors are bootstrap standard errors across model fits (N = 50 iterations). **E)** Histogram of changes in bout frequency induced by the thermosensory receptive fields fit on the Plaid (blue) and Replay (red) conditions. These are the expected changes in bouts per second between the most suppressive and most activating stimuli according to the receptive fields. Statistics are based on a boot-strapped KS test see Methods. p<0.001; statistic = 0.1245; n = 104. **F)** Scatter plot of the standard deviation in bout-frequency of the thermosensory receptive field effects in each of 52 experiments when using the Replay receptive fields versus using the Plaid receptive field (purple dots). Dashed line is identity, any dot below the line signifies greater modulation by the Plaid than Replay receptive field. Wilcoxon signed rank test p<0.001; statistic = 0.0; N = 104.

temperature change. The receptive field describing the influence of swim history on swim generation showed a sharper transition out of the refractory period 300 ms before swim initiation in the Plaid condition (Fig 3D). Specifically, swim generation was significantly more suppressed for interbout intervals below 100 ms and significantly more enhanced for interbout intervals between 400–600 ms in the presence of sensorimotor feedback (S2B Fig). Just like the increased predictability of swims during the Plaid condition (Figs 3B and S1D) this suggests that behavior was more structured in the presence of sensorimotor feedback.

The increased magnitude of coefficients in the thermosensory receptive field during Plaid conditions suggested a stronger modulation of behavior by thermal stimuli. To test if this was the case during our experimental conditions, we used the thermosensory receptive field to determine the predicted modulation in swim frequency mediated by the stimulus. Since the receptive field measures how much the stimulus affects the probability of swim bouts, we used the Plaid and Replay receptive fields to compute the swim probability at each time point in

each experiment. This allowed us to determine the change from the average bout frequency mediated by the thermosensory receptive field across time. As expected, the stimuli experienced by larval zebrafish during the experiments were predicted to have a stronger influence on behavior in the presence of sensorimotor feedback than in the Replay condition. This can be seen in the overall distribution of changes in bout frequency (Fig 3E). Here, a wider distribution signifies that the stimulus induces a larger range of bout frequencies between times where the stimulus is opposed to the receptive field and times where it is aligned with the receptive field in the Plaid compared to the Replay condition. In line with this observation, the standard deviation of stimulus bout frequency modulation across time induced by the receptive fields is larger for the Plaid than Replay receptive field across all experimental stimuli (Fig 3F). The overall effects of the sensory receptive field on bout frequency were small, but consistent with the expected modulation given the limited thermal range of $3\,°C$ we tested (see Limitations below). In summary, these results suggest that sensorimotor feedback enhances the influence of the thermal stimulus on behavior generation.

## Mixed selectivity neurons represent sensorimotor feedback

We previously identified neurons across the larval zebrafish brain that jointly encode thermosensory stimuli and information about generated behavior [19]. The neurons could therefore encode information about sensorimotor feedback, e.g., by specifically capturing coincidences of temperature change and behavioral actions. To test this idea, we tested whether a linear model could classify Plaid versus Replay experiments based on the activity of the mixed selectivity neurons (Fig 4A). This follows the notion that important information is often encoded such that it can be linearly decoded by downstream neural circuits [38–40]. Since we didn't record neural activity during our behavioral experiments we made use of the predictive power of previously fit CNN models [19] to generate predictions of how the mixed selectivity neurons would have responded during the 52 Plaid and Replay experiments. We fed the stimuli as well as elicited behaviors (Fig 4B) as inputs into CNN models we previously fit when we identified mixed selectivity neurons [19]. Each model then predicts the calcium response of an individual mixed selectivity neuron. These models were fit under open-loop conditions (akin to the Replay condition). Therefore in the context of the closed-loop Plaid condition, naturalistic neural representations may not be perfectly recapitulated. However, analysis of these neurons suggested that their responses depend on the coincidence of temperature change and behavioral responses, which occurred during the imaging paradigm [19]. This suggests that the responses we identified approximate the true physiological tuning of these neurons which can therefore be replicated by the CNN model. We computed the responses of 1023 nonlinear mixed selectivity neurons across 100 random train/test splits of the behavioral data (Fig 4C).

In Costabile et al., 2023, we found that neuronal subtypes were broadly represented in all animals [19]. Consequently, we performed PCA on the complete set of 1023 neurons to reflect the average activity space characteristic of all fish, rather than the precise neuronal composition of any single individual. We then trained a linear classifier on the first 10 principal components across the neural predictions. This classifier therefore answers the question whether the Plaid or Replay condition can be linearly decoded from neural activity in mixed selectivity neurons. Notably, our CNN models represent intervening processing between the inputs (temperature stimulus and behavior) and the activity of mixed selectivity neurons. This nonlinear processing could either make it easier to decode the presence or absence of sensorimotor feedback or obscure this information. In the latter case we would expect that the presence or absence of sensorimotor feedback is easier to decode from the inputs than from

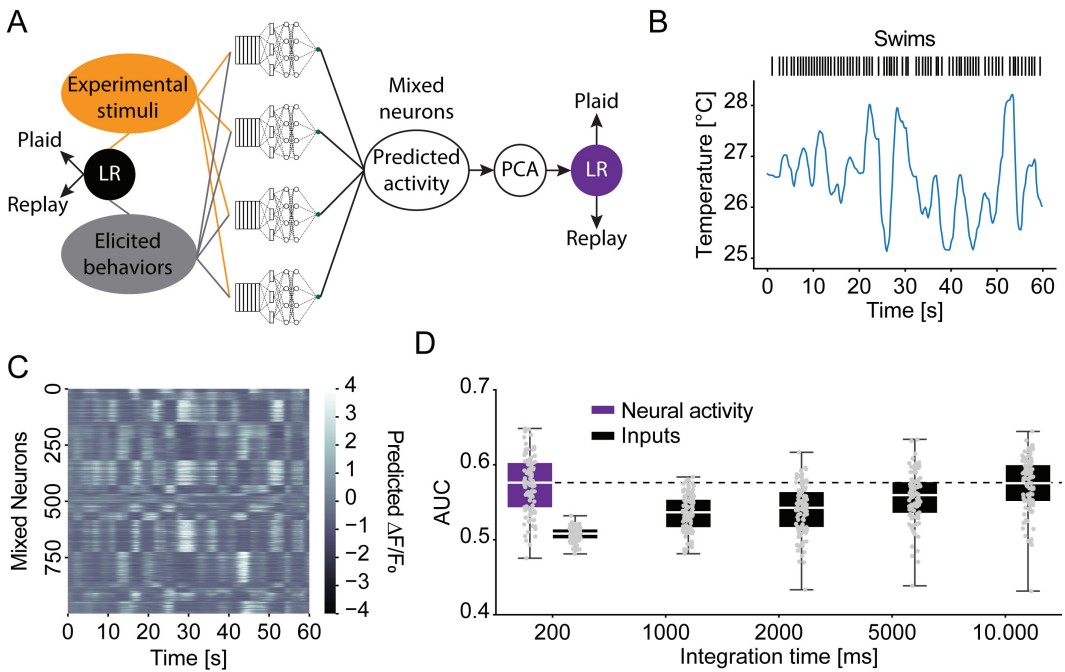

**Fig 4. Mixed-selectivity neurons provide a substrate for identifying sensorimotor feedback. A)** Schematic of the classifier approach, showing how stimuli and behaviors are fed through mixed-selectivity neuron CNN models on the one hand and directly used to classify Plaid and Replay conditions on the other. **B)** Example stimulus and swim behavior during a 60 s period of a randomly selected experiment. **C)** Corresponding predicted calcium activity of nonlinear mixed-selectivity neurons, clustered according to response correlation for display purposes. **D)** Classifier performance (as ROC-AUC) of a logistic regression based classifier trained on predicted neural activity (purple) or the inputs across the listed integration times. Dashed line indicates median performance of neural classifier. (N = 100 random train/test splits).

the neural activity. Therefore, for comparison, we trained an equivalent linear classifier on the inputs themselves, i.e., the stimulus and the generated behavior. Comparing the performance of the two classifiers then tested whether mixed selectivity neurons represented the information in a manner that is easier to decode by a linear classifier than a classifier operating on the raw inputs as encoded in sensory temperature neurons and motor neurons respectively.

A linear classifier trained on the activity of mixed selectivity neurons was better than chance level in separating Plaid and Replay experiments, indicating that these neurons carry information relevant to detecting sensorimotor feedback (Fig 4D). The comparison classifier trained on input data however failed to match the performance of the neuron classifier unless integration time was extended to 10 s. Importantly, this input classifier was trained to optimally integrate the sensory and behavior information across the 10 s. This argues that mixed selectivity neurons optimally integrate information about stimuli and behavior across time to allow for the detection of sensorimotor feedback by downstream circuits.

## Discussion

Here we identified an intriguing modulation of zebrafish sensorimotor transformations in the context of thermal stimuli. When changes in temperature are coupled to the behavior of the fish (sensorimotor feedback), the influence of the temperature stimulus on swim-bout generation is greater than in the absence of this feedback. We furthermore pinpoint a possible neural substrate informing the fish about the coupling between behavior and stimuli. Namely,

mixed-selectivity neurons we previously identified [19] encode information about temperature and ongoing behavior in a manner that allows a linear classifier to distinguish between the presence and absence of sensorimotor feedback.

Our motivation for the current study was to understand how context influences sensory processing in thermoregulation. When exploring their environment, animals need to appropriately categorize sensory cues to optimize their behavioral actions. This includes information on whether a sensory change was caused by the animal or the environment. In some contexts, self-generated sensory feedback is a distractor and actively suppressed by efference copy mechanisms [41–44]. In other cases, however, such as during active sensing, sensory feedback of one's own actions is actively sought [45–47]. Under some conditions, the absence of sensory feedback signals futile actions to an animal which are subsequently suppressed [33,48, 49]. This indicates a broad modulation of behavior according to sensorimotor feedback.

Larval zebrafish are ectotherms and navigate temperature gradients to thermoregulate [10–12]. Within a thermal gradient, temperature changes are tightly coupled to behavior, since each swim changes the location of the animal within the gradient. On the other hand, changes in irradiation (e.g., through changes in cloud cover) will also lead to temperature changes within the water, especially since zebrafish frequently inhabit shallow pools [50]. We therefore hypothesized that larval zebrafish might evaluate whether temperature changes are the result of their own actions in order to adjust their behavioral responses. This would allow them to specifically engage navigation behavior in the presence of a thermal gradient. To test this hypothesis, we used a paradigm that allowed us to variably couple or decouple thermal stimuli from the location of the fish while keeping the temporal sequence of temperatures the same. This allowed us to compare the processing of the same stimuli under two conditions, one with full sensorimotor feedback, the other in its absence. To characterize the processing, we modified a technique based on convolutional neural networks that allowed us to efficiently extract receptive fields from data without requiring white-noise stimuli as inputs [19]. These receptive fields compactly represent which thermosensory features drive swim generation and how successive swims influence each other. We previously extracted similar receptive fields using white-noise stimuli, i.e., in the absence of sensorimotor feedback [23]. The receptive fields we now identified look qualitatively similar, albeit less compact, likely due to the much larger autocorrelation time of stimuli during natural swimming. Notably, sensorimotor feedback led to significant changes in these receptive fields (Figs 3C, 3D and S2). The thermosensory receptive field shows a clear dependence of swim generation on the derivative of the temperature stimulus; both increases in temperature and changes in the speed of temperature change influence swim generation. These features are strongly enhanced during the Plaid condition (sensorimotor feedback), indicating that integration of thermosensory information and motor information enhances responses to the stimulus. This is reflected in the fact that the Plaid receptive field induces a stronger modulation of swim frequency than the Replay receptive field (Fig 3E and 3F). At the same time, each swim is followed by a refractory period. Sensorimotor feedback enhances this effect by sharpening the transition from suppression for 300 ms after the last swim to enhancement for longer delays (Fig 3D). This suggests a higher regularity in swim generation during the Plaid condition, which may also be the reason why models trained on the Plaid condition generalize better than those trained on the Replay condition (Fig 3B). Taken together, this indicates that larval zebrafish increase the influence of the stimulus on behavior and the regularity of swim intervals when in a thermal gradient where behavior can be productively used for thermoregulation.

Behavioral thermoregulation is prevalent across motile organisms from bacteria to humans [5,13,15,51–54]. In the presence of thermal gradients, *E. coli* [55], C. elegans [5,6,51], Drosophila [52,56,57], fish [10–12,23,58,59] and mammals [14,60–65] will seek out preferred

temperatures. At the same time, both fish and mammals have been shown to learn motivated behaviors, such as lever presses, to control the temperature of their environment [20,66–68]. This indicates that animals will enact thermoregulatory strategies that are appropriate in the given situation. How they perform these adjustments, however, is unclear. Here we suggest that at least in the case of larval zebrafish, mixed selectivity neurons, which integrate thermosensory and behavioral information, form a neural substrate for switches in thermoregulatory strategies. We use CNN models previously fitted on the mixed selectivity neurons [19] to predict their respective activity during the Plaid and Replay experiments. A classifier trained on this neural activity can separate Plaid and Replay conditions, similar to a classifier trained to ideally integrate sensory and behavioral information over ten seconds. This suggests that mixed selectivity neurons integrate thermosensory and behavioral information in a manner that allows larval zebrafish to decide whether their environment contains a navigable thermal gradient. This subsequently biases their behavior by adjusting sensorimotor computations to a mode in which actions are more stimulus driven. These findings suggest that larval zebrafish infer the presence of a gradient from the relationship of temporal temperature dynamics with behavior rather than specific spatial patterns. This mirrors natural ecology where fish infer spatial thermal gradients through temporal sequences generated by navigation [69] effectively using time to reconstruct space.

## Limitations

Here we demonstrate that larval zebrafish adjust the processing of thermal stimuli when their own behavioral actions control the temperature they experience. We suggest that this allows the animal to optimize behavioral output for thermal gradient navigation and hence thermoregulation. To directly address this point, it would be highly desirable to fit a model on the data that explains not only swim generation but also swim kinematics. This would allow to directly compare gradient navigation abilities of models that process stimuli as the fish does in the Plaid versus Replay condition. If the model fit on the Plaid condition would overall remain closer to the preferred temperature, it would strongly suggest that the changes we observe indeed optimize sensorimotor transformations for gradient navigation if there is sensorimotor feedback. Unfortunately, technical limitations prevent us from probing a large enough temperature range to make such a comparison feasible. Our thermal stimuli span a 3 °C range around the preferred temperature of larval zebrafish ($\sim$ 26 °C). While this is a large enough range to observe modulation in swim generation frequency (as modeled here), kinematic parameters such as swim distance barely vary within this range. In line with this, an attempt to fit a model that predicts swim distance was unsuccessful. Furthermore, the temperature range is not large enough to model thermal navigation. So even if we could fit models for all swim features, we still would not be able to use them for simulations of thermal navigation. These would require gradients in the 20–30 °C range [11,12,23] and there would be no guarantee that our current models would generalize over this temperature range.

We currently infer the importance of mixed selectivity neurons through indirect means. In a previous study a CNN model predicted that mixed selectivity neurons encode integrated sensorimotor information, specifically the coincidence of temperature change and behavior. Since these CNN models have high predictive power, we used them here to infer the response of these neurons during Plaid and Replay behavior. However, direct neural recordings during virtual navigation could clarify how these neurons represent sensory motor feedback. Combining functional calcium imaging with a head-embedded version of the thermal plaid paradigm could reveal whether these neurons selectively enhance sensitivity to self initiated

feedback. These experiments would directly link the computational role to underlying circuit mechanisms that support adaptive thermoregulatory behavior.

## Materials and methods

### Ethics statement

Animal handling and experimental procedures were approved by the Ohio State University Institutional Animal Care and Use Committee (IACUC Protocol #: 2019A00000137-R1).

### Fish strains

All experiments were performed in pigmented offspring of incrosses between mitfa +/-; Elavl3-H2B:GCaMP6s [22] animals.

### Behavioral setup and temperature calibration

The behavioral setup used was as described previously [23]. Due to slight modifications in the equipment used, we describe the components again below.

While fish were freely exploring a circular arena with a diameter of 100 mm and a depth of 4 mm, we acquired images at 250 Hz using a Mikrotron 1362 camera (SVS-Vistek GmbH, Germany) utilizing a NI PCIe-1433 frame grabber (National Instruments Corporation, USA). The arena was illuminated from below using an array of 880 nm IR LEDs. Visible light and reflections of the laser were blocked using a combination of three filters: A 25 mm diameter 900 nm shortpass filter (Thorlabs, USA), a 50 mm diameter 900 nm shortpass filter (Edmund Optics, USA) and a 50 mm diameter 750 nm longpass filter (Edmund Optics, USA). Custom written software in C# (Microsoft, USA) extracted the fish position and heading angle in realtime (average time from image acquisition to position: 0.3 ms). The position information was used to send voltage commands via an NI PCIe-6323 DAQ board (National Instruments Corporation, USA) to a set of 6210H Galvos rotating 3 mm diameter X/Y scan mirrors (Cambridge Technology, USA). At the same time the output power of an SDL-980-LM-8000T (Shanghai Dream Lasers, China) infrared laser operating at 980 nm with a maximum output power of 8 W was controlled according to the behavioral paradigm by supplying appropriate voltage commands to the laser current driver. The laser beam was cleaned by first focusing the beam onto a pinhole using a 50 mm focal length lens (Thorlabs, USA) and subsequently collimating using another 50 mm focal length lens (Thorlabs, USA). The beam was then slightly focused using a 750 mm focal length lens (Thorlabs, USA) to a spot diameter of 5 mm at sample, measured using an IR fluorescent alignment disc (Thorlabs, USA).

Temperature calibration was performed as described in [23].

### Experimental paradigm

Each experiment was performed in a different fish. Compared to running the Plaid and Replay paradigm in the same fish this had the disadvantage of introducing more noise due to behavioral variability between fish. This choice was made because otherwise the Replay experiments would always need to be run in fish that already went through the Plaid paradigm. In other words, the results could have been confounded by longer-term habituation effects. A "thermal plaid" was presented to the fish rather than a circular gradient, to better decouple the relationship between the temperature experienced by the fish and the distance to the edge of the arena. This was done to mitigate confounds by thigmotaxis behavior in which larval zebrafish track edges. Having multiple mini-gradients allowed excluding all data close to the edge from the analysis (see below).

In the Plaid condition the laser power at sample, and therefore temperature, delivered to the fish was determined by the position within the arena as follows:

$$P_{mW} = 750\,\text{mW}\,(0.5\sin(2\pi\mathbf{x}/15) + 0.5\sin(2\pi\mathbf{y}/15)) + 1250\,\text{mW} \tag{1}$$

where *x* and *y* are the fish centroid coordinates in mm from the top left corner.

This led to a plaid with a period of 21 mm and laser powers at sample ranging from 500 mW to 2000 mW, effectively presenting multiple mini-gradients to the fish. For each fish run in the Plaid condition, one corresponding fish was run in the Replay condition. This fish got an exact copy of the laser powers (and hence temperatures) delivered to the Plaid fish.

Each fish was first habituated to the chamber for 10 minutes and then subjected to a baseline condition with constant power of 500 mW for 20 min followed either by the Plaid or the Replay condition for another 20 min.

## Data analysis

All data analysis was performed in Python using Tensorflow [24] and scikit-learn [25].

**Swim bout identification.** During acquisition a small image containing the tracked fish and background was saved at each frame. These images were used to extract the tail and torso of the fish frame-by-frame by skeletonizing the fish. These data were used to determine the heading of fish within the arena and to calculate the cumulative tail bend angle ("Tail angle"). The standard deviation in a sliding window (of size 10 frames) was subsequently computed on the Tail angle ("Swim vigor", [26]). Whenever this metric crossed an empirically set threshold of 0.1 radians/frame the start of a swim bout was detected and the end of the swim was determined by the metric falling below threshold. Importantly, this approach allowed to detect both in-place turns as well as swims leading to displacement of the fish centroid. Swim kinematics, such as the net displacement of the fish centroid ("Displacement") and heading change ("Turn angle") of each swim were subsequently extracted. The Displacement was defined as the euclidean distance between the average position in the five frames before the bout start and the average position in the five frames after bout end. Similarly, the Turn angle was defined as the difference in heading angles between the average five frames before bout start (calculated as an average vector) and the average five frames after bout end. As this leaves ambiguity with respect to direction, it was decided that the smallest angle between the start and end angles would constitute the turn. Analysis was subsequently limited to those swim-bouts that occured at least at a distance of 4 mm from the edge to avoid confounds caused by the edge limiting the possible movement repertoire of the larvae.

Relevant Python file in repository: *processing.py*

**Network model and training.** The network model was similar to the models used in [19]. A simple model-architecture with one convolutional layer (made up of 20 units) and two deep layers with 64 units each was kept. As in [19] the convolutional layer was linear, while "Swish" [27] was used as the activation function of the dense layers, as a continuously differentiable alternative to ReLu. Dropout [28] was used after each layer to aid generalization. Instead of predicting a continuous output variable, the goal of the network however was to classify outputs into swim-bout and non-swim-bout frames. The output layer was linear and trained to approximate the log-probability of the occurrence of a swim-bout. To this end, binary cross-entropy was used as the loss-function during training. We note that the architecture of the model and the chosen loss-function were not optimized. However, a separate dataset was used to optimize the weight decay (S1 FigB) hyperparameter and the number of training epochs (S1 FigC).

For the analysis presented in the paper, networks were trained on all Plaid or all Replay data (combination across 52 separate experiments in each group). This fit was repeated 50 times each to estimate the distribution of solutions found by the networks. This approach was chosen, since the data from one experiment was not sufficient to train the models and there was clear variability across training indicating the presence of multiple local minima.

To generate control data for receptive field extraction the model outputs (bout starts) were circularly permuted relative to the inputs (temperature stimulus, swim bout history). The circular shift was fixed at 1/3 of the 20 minute long Plaid/Replay period.

Relevant Python files in repository: *model_defs.py, utility.py, fit_models.py*

**Receptive field extraction and effects.** Linear receptive fields were extracted as described previously. Specifically, we calculated the derivative of the output of the network (the log probability of emitting a swim-bout) with respect to its inputs (10 s of sensory history and 10 s of previous swim bout ends), which is equivalent to the extraction of a spike-triggered average [19]. We note that the receptive fields show the influence of the inputs on the log-probability of emitting a swim-bout - the effect on probability is non-linear.

To assess the effect of the thermosensory receptive fields on bout frequency, the Plaid and Replay receptive fields $\vec{k}$ were correlated point-by-point with the stimulus presented in each of the 52 experiments (since each Plaid/Replay pair received the same stimulus). Since the receptive field was extracted via Taylor expansion, its effect encodes the change in log-probability relative to the average log-probability $\bar{p}$ of producing a swim. At each point the average log-probability was added and the overall swim-probability was calculated using a logistic transform:

$$\Delta lp(t) = \vec{k}^T \vec{s}(t) \tag{2}$$

$$\bar{lp} = \log\left(\frac{\bar{p}}{1 - \bar{p}}\right) \tag{3}$$

$$p_{bout}(t) = \frac{1}{e^{-1 lp(t) - \bar{lp}} + 1} - \bar{p} \tag{4}$$

$$bf_{Hz}(t) = p_{bout}(t) 25 s^{-1} \tag{5}$$

The resulting probabilities were used to calculate a histogram of swim bout probability modulation around the mean for the Plaid and Replay receptive fields. At the same time, the receptive field effects for each of the 52 stimuli were determined as the standard deviation of bout probabilities induced by the receptive field.

Relevant Python files in repository: *utility.py, rf_analysis.py*

**Neuron models and classifier.** To test the ability of mixed selectivity neurons [19] to classify the Plaid and Replay conditions, previously fit CNN models [29,30] were used to convert temperature stimuli and behaviors performed by larval zebrafish during the experiments into predicted neural activity. To this end, for each experiment behavioral features (swim starts, swim displacements and turn angles) as well as the temperature stimulus were binned to 5 Hz, the frequency at which the MINE models were fit. Subsequently all non-linear mixed-selectivity neurons were selected from [19] and the behavioral and stimulus data was fed into the models as predictors to generate likely calcium activity within these neurons during the behavioral experiments. The choice to focus on the nonlinear mixed-selectivity neurons was arbitrary, however it reduced the amount of data and the idea was that these might capture the most relevant aspects of stimulus-behavior integration.

The dimensionality of the predicted neural activity was subsequently reduced using principal component analysis, retaining the first 10 components explaining 91% of the total variance. A logistic regression model was subsequently trained on the data with the goal of classifying Plaid vs. Replay experiments (11 total parameters). Two thirds of the experiments were used to train the classifier with a five-fold split of the data being used to optimize a ridge penalty in the model. One third of experiments was used as a test set to assess the performance of the classifier. This was repeated across 100 random train/test splits.

As a comparison, a logistic classifier was trained directly on the inputs to the neurons, the temperature as well as behavior values. This was done using either the inputs only at the current time point (five total parameters) up to using inputs for the last 10 seconds up to the current time point (201 total parameters), to assess how integrating across time would aid the classification.

Relevant Python files in repository: *safe_virtres_input_data.py, virtres_analysis.py*

**Statistics.** Except where stated in the figure legend, bootstrap standard errors were reported for all quantities. Where significance was tested, nonparametric tests were performed. Non-parametric tests were chosen, since the underlying data for the quantities come by definition from constrained ranges and therefore cannot be normally distributed. To compare distributions, the KS (Kolmogorov-Smirnov) statistic was bootstrapped. This approach was chosen because most of the variability arises between fish rather than between individual swim bouts. However, using a standard KS test would evaluate the statistic with respect to the number of individual swim bouts. The significance was therefore calculated using a bootstrap test as briefly outlined in the following. The KS statistic was calculated for the true sample comparison. For bootstrapping, all fish data were combined and bootstrap variates were generated by drawing data with replacement from this pool generating two samples with the same number of fish in each as in the original comparison. The KS statistic across these two samples was subsequently calculated. Repeating this procedure 10,000 times generated a baseline distribution. The p-value was subsequently calculated as the fraction of KS statistics in the baseline distribution that were at least the same value as the KS statistic of the true comparison [31].

## Supporting information

**S1 Fig. Extended paradigm and network optimization data. A)** Stimulus autocorrelation across the 52 Plaid experiments (Replay received same stimulus). Shaded region indicates bootstrap standard error across the 52 experiments. **B)** Classifier performance as area under the ROC curve when fitting the CNN models with different $l_2$ penalties (weight decay). $10^{-5}$ was chosen as the final penalty. Each dot is a separate fit, $N = 5$ fits. **C)** Same as B) but for differing amounts of training epochs. 100 training epochs were chosen as the final number. **D)** QQ-Plot of the true proportion of swim-bouts within experimental frames binned based on the model-predicted probability. Overlap with the identity line (dashed) would indicate perfect prediction. Shaded areas are bootstrap standard error across 50 separate fits. (TIF)

**S2 Fig. Extended paradigm and network optimization data. A)** Blue dots: For each time point the p-value obtained from a ranksum test comparing the temperature receptive field values of the Plaid and Replay condition (N = 100 model fits). The black lines show p <0.05; p <0.01; p <0.001 significance level after correcting for multiple comparison across 25 time points. **B)** Blue dots: For each time point the p-value obtained from a ranksum test comparing the bout history receptive field values of the Plaid and Replay condition (N = 100 model fits).

The black lines show <0.05; p <0.01; p <0.001 significance level after correcting for multiple comparison across 25 time points.
(TIF)

## Acknowledgments

We thank Danica Matovic and Bradley Cutler for valuable comments on the manuscript.

## Author contributions

**Conceptualization:** Martin Haesemeyer.

**Data curation:** Martin Haesemeyer.

**Formal analysis:** Lindsay S. Anderson, Martin Haesemeyer.

**Funding acquisition:** Martin Haesemeyer.

**Investigation:** Lindsay S. Anderson, Sina Schwinn, Delia Calderon.

**Methodology:** Martin Haesemeyer.

**Software:** Jamie D. Costabile, Martin Haesemeyer.

**Supervision:** Martin Haesemeyer.

**Validation:** Lindsay S. Anderson.

**Writing – original draft:** Martin Haesemeyer.

**Writing – review & editing:** Lindsay S. Anderson.

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
