## [Editor Report · Decision Letter 0]

16 Nov 2024

PCOMPBIOL-D-24-01934

Sensorimotor integration enhances temperature stimulus processing

PLOS Computational Biology

Dear Dr. Haesemeyer,

Thank you for submitting your manuscript to PLOS Computational Biology. As with all papers, your manuscript was reviewed by members of the editorial board. Based on our assessment, we have decided that the work does not meet our criteria for publication and will therefore be rejected. 

Your analysis of thermoregulatory behaviour is very interesting. However the computational aspect has already been presented in your previous publication Costabile, J. D., Balakrishnan, K. A., Schwinn, S. &

Haesemeyer, M. 2023 (n.b. the bibliography is missing the journal and year references), and thus the current manuscript does not meet the journal's criteria for novelty.

We are sorry that we cannot be more positive on this occasion. We very much appreciate your wish to present your work in one of PLOS's Open Access publications. Thank you for your support, and we hope that you will consider PLOS Computational Biology for other submissions in the future.

Yours sincerely,

Lyle Graham

Section Editor

PLOS Computational Biology

Feilim Mac Gabhann

Editor-in-Chief

PLOS Computational Biology

Jason Papin

Editor-in-Chief

PLOS Computational Biology
---

## [Decision Letter · Decision Letter 1]

27 Mar 2025

PCOMPBIOL-D-24-01934R1

Sensorimotor integration enhances temperature stimulus processing

PLOS Computational Biology

Dear Dr. Haesemeyer,

Thank you for submitting your manuscript to PLOS Computational Biology. After careful consideration, we feel that it has merit but does not fully meet PLOS Computational Biology's publication criteria as it currently stands. Therefore, we invite you to submit a revised version of the manuscript that addresses the points raised during the review process.

Please submit your revised manuscript within 60 days May 27 2025 11:59PM. If you will need more time than this to complete your revisions, please reply to this message or contact the journal office at ploscompbiol@plos.org. Please include the following items when submitting your revised manuscript:

We look forward to receiving your revised manuscript.

Kind regards,

Matthieu Louis

Academic Editor

PLOS Computational Biology

Lyle Graham

Section Editor

PLOS Computational Biology

**Journal Requirements:**

1) Please provide an Author Summary. This should appear in your manuscript between the Abstract (if applicable) and the Introduction, and should be 150-200 words long. The aim should be to make your findings accessible to a wide audience that includes both scientists and non-scientists. Sample summaries can be found on our website under Submission Guidelines:

**Reviewers' comments:**

Reviewer's Responses to Questions

**Comments to the Authors:**

**Please note that one of the reviews is uploaded as an attachment.**

Reviewer #1: Comments have been uploaded as an attachment.

Reviewer #2: The authors of 'Sensorimotor integration enhances temperature stimulus processing' investigate if behavioral feedback coupled to changes in sensory input allow for more effective temperature control. Using a behavioral paradigm, they first demonstrate that an animal that navigates in a spatial environment with a structured temperature pattern is more effective at reducing its temperature experience, compared to an animal where movement and stimulus are uncoupled. I overall found this paper a compact, well-described observation with the computational aspects aiding to suggest potential mechanisms for the observation. However, I have a few general comments regarding a few statements and comparisions that if adressed would strengthen the results.

General comments:

The effect size of Fig. 1 E seems rather small. It would be useful to put this result in context - what are the changes in temperature across the 'plaid' and what percentage does the 0.1 degree Celsius represent? (This comes later in the 'Limitations') but it would help the reader to have context here.

In Figure 2, the authors suggest that the overall behavior in stimulus-coupled and uncoupled animals is similar. They begin their argument with learned helplessness, an effect previously described for visual paradigms. They then dismiss this by showing that gross behavioral statistics are similar. Here, proof by graphical overlay is used, which in itself is not bad, however, I'm suprised the peak in Fig. 2A is not significant. Please run an appropriate statistic (for example a KS-test). The graphing of this information would benefit from either using linegraphs, or opacity to show both histograms more clearly.

In addition, I wonder if learned helplessness would not show up over time, rather than in overall statistics. However, this is in general a side-note and I don't think this aspect is central to the paper, and I was generally a bit distracted by this Figure. In general, the latter part of the paper argues that swim bouts are different (e.g. Fig. 3E) so this line of reasoning should be clarified in the text.

The authors then use a previously published CNN-based approach to determine receptive fields from non-white noise (i.e., highly correlated) stimulus data. The paper would be easier to read if the authors added a (minimal) description of this model is and how it works. White-noise based approaches are widespread, as they have some neat statistical properties. It would be useful to state why this alternate approach works.

The final section concerns inference of mixed selectivity neurons on behavior, with the rationale that these neurons may encode relevant information such that it is easily decodable later on. As this paper is using only behavioral data, a previously used generative model supplied neuronal traces which were then fed into a classifier to predict bouts. This model is compared to a linear classifier and performs better. However, this comparison is flawed: The neuron model has access to non-linear features extracted from the data, wheras the linear classifier is restricted to linear features. It is very obvious that this would be a mismatched comparison. A better comparison could be a non-linear classifier, a random forrest or at least a random weight CNN with similar layers.

Specific line edits:

Ln 54: This sentence sounds like editorializing and is currently not supported by references. I suggest reformulating and providing evidence.

Fig. 3B, caption: 'rotated control data' - do you mean circularly permuted? Shifted? Or indeed a spatial rotation relative to the plaid? This phrase is otherwise unfamiliar to me.

Minor:

Figure 1:

- formatting of superscripts in the caption

Figure 3B: formatting of p-value and statistics

-Ln.151 Are actively sought -> is actively sought

- Ln 184 italicize species

Reviewer #3: Overall I like the paper -- it's fun to read and interesting, and gets at some important issues in behavioral neuroscience about how information is incorporated by animals when making behavioral decisions. I would appreciate some aspects being discussed a bit more systematically, and a have a basic question about the framing of the main result, but otherwise the comments are relatively minor.

My broader comment is about framing. The fish are described as "actively controlling" the temperature because there is a direct connection between where they move and what temperature stimulus they experience. To me this seems like the default in a ecological setting, and would also be true for terrestrial animals moving around on a spatial thermal gradient, for example. Unless the fish are making spatial maps and storing that information in their brain (are they? I don't recall it coming up), wouldn't ANY movement appear TO THE FISH as the same thing, regardless of how temperature was changing with time? I understand that in Replay conditions the fish are not moving in a virtual plaid gradient anymore, but they do still have the temperature change as they move. It seems like the main difference between Plaid and Replay is what happens when the fish is sitting still -- in Replay they might experience temperature changes, but never in Plaid. Could another way to talk about the difference in the experiments is that in one type their stationary periods are being interrupted and in the other they are not? From the picture in Fig. 1C it seems like navigating that environment would provide plenty of sensory feedback even if kind of scrambled. I'm not asking to change how you talk about everything, but it might be useful to acknowledge what happens differently from the fish's point of view. For them every stimulus is temporal.

It seems here that the main result of the paper is that when the temperature changes during bouts, the fish pay behavior is more influenced/controlled by the stimulus. If that's true, wouldn't it not be very important to have any kind of coherent spatial pattern? If you change the temperature every time the fish moved, would you expect similar results? Or similarly, intentionally giving fish changing temperatures while they are at rest but NOT when they are in bouts would minimize the relationship between behavior and stimulus.

More specific comments:

* The Introduction is really short, and most of it isn't background, but it's the abstract material repeated and then talking about the experiments in the manuscript itself. Setting a broader context would be helpful I think. Actually I think the examples in line 155ff and 183ff would fit nicely in the Introduction.

* What percentage of their time do the fish spend in bouts? Are they traces in Fig. 1D fairly representative, or are they unusual because of the changing temperature stimulus? It would be helpful to have a baseline idea.

* The apparatus is very cool, but I think it would help if the reader understood better why it was important/necessary to build it. For example, for a land-crawling creature a fixed spatial gradient could accomplish the same thing as Plaid -- is being in water what would make that difficult?

* (~line 60) There is a clear comparison with Plaid/Replay -- could there be some comment or data shown what the fish's baseline behavior is like in an isotropic environment as well?

* (line 107) I couldn't tell what precisely delta_f means. It says changes in bout frequency -- does this mean between subsequent bouts? Or before vs. after the stimulus shows up? And why does having a wider distribution of this quantity mean there is a stronger influence from the stimulus on behavior? This might be addressed in other work, but I feel I'm missing some of the logic in the manuscript text.

* (line 109) Maybe rewrite this sentence for clarity, it's a little clunky.

* (line ~285) The turn angle means the drift in direction during a bout? Why does it change so much during the bout (seen in Fig. 1D) -- and why does it have noisy bursts like that?

* In Fig. 1, it would be really nice to see the Plaid pattern (and the other one) larger, and perhaps with a real trajectory overlaid? (either the one from 1D or another representative one). It would be a nice visual and would also give a better sense of the scale of bout distances compared to the gradient size.

* Reference [15] might have a mistake (volume 0?)

* (line 184) I think most of those genus/species should be in italics.

* I'm not clear on what a more "structured" output is? The term is used a lot and I'm not sure of the precise meaning.

* The limitations section is pretty light -- the whole study has only one limitation? Some more thought on what could be done to expand the scope or test things in additional ways would be appreciated. Measuring neural activity for example?

**Have the authors made all data and (if applicable) computational code underlying the findings in their manuscript fully available?**

Reviewer #1: Yes

Reviewer #2: Yes

Reviewer #3: Yes

PLOS authors have the option to publish the peer review history of their article (what does this mean?). If published, this will include your full peer review and any attached files.

Reviewer #1: No

Reviewer #2: No

Reviewer #3: **Yes: **Mason Klein

**Figure resubmission:**
---

## [Decision Letter · Decision Letter 2]

13 May 2025

Dear Dr. Haesemeyer,

We are pleased to inform you that your manuscript 'Sensorimotor integration enhances temperature stimulus processing' has been provisionally accepted for publication in PLOS Computational Biology.

Best regards,

Matthieu Louis

Academic Editor

PLOS Computational Biology

Lyle Graham

Section Editor

PLOS Computational Biology

Reviewer's Responses to Questions

**Comments to the Authors:**

Reviewer #1: The revised manuscript satisfactorily addresses all of my prior concerns.

Reviewer #2: The authors have fully addressed my prior comments. The new edits substantially improve the clarity of the paper, especially by clearly explaining the main finding and providing reasons for the chosen conditions. The added explanations provide more context to the models and clarify the results regarding the linear decoders. I recommend acceptance.

Reviewer #3: The authors have addressed the concerns I had, and I think it's a really nice paper. Thank you for all the work you put into revising.

**Have the authors made all data and (if applicable) computational code underlying the findings in their manuscript fully available?**

Reviewer #1: Yes

Reviewer #2: None

Reviewer #3: Yes

PLOS authors have the option to publish the peer review history of their article (what does this mean?). If published, this will include your full peer review and any attached files.

Reviewer #1: No

Reviewer #2: No

Reviewer #3: **Yes: **Mason Klein

---

## [Editor Report · Acceptance letter]

PCOMPBIOL-D-24-01934R2

Sensorimotor integration enhances temperature stimulus processing

Dear Dr Haesemeyer,

I am pleased to inform you that your manuscript has been formally accepted for publication in PLOS Computational Biology. Your manuscript is now with our production department and you will be notified of the publication date in due course.

With kind regards,

Judit Kozma
